Wolf spider burrows from a modern saline sandflat in central Argentina: morphology, taphonomy and clues for recognition of fossil examples

Mendoza Belmontes Fatima famebel@exactas.unlpam.edu.ar fa.belmontes@hotmail.com 1
Melchor Ricardo N. 2
Piacentini Luis N. 3
1 UNLPam, FONCyT doctoral scholar , Santa Rosa , La Pampa , Argentina
2 INCITAP- UNLPam, CONICET , Santa Rosa , La Pampa , Argentina
3 Museo Argentino de Ciencias Naturales ‘Bernardino Rivadavia’, CONICET , Buenos Aires , Argentina
Piñeiro Graciela
Electronic publication date: 2018 Jun 29
Publication date: 2018
Volume: 6
Electronic Location ID: e5054
Received 2018 Mar 9; Accepted 2018 Jun 2
Copyright: ©2018 Mendoza Belmontes et al.
Copyright year: 2018
Copyright holder: Mendoza Belmontes et al.
License: This is an open access article distributed under the terms of the Creative Commons Attribution License, which permits unrestricted use, distribution, reproduction and adaptation in any medium and for any purpose provided that it is properly attributed. For attribution, the original author(s), title, publication source (PeerJ) and either DOI or URL of the article must be cited.
License URL: https://creativecommons.org/licenses/by/4.0/

Keywords: Spider burrow, Wolf spiders, Neoichnology, Predation, Burrow reoccupation, Burrow modifications, Saline lake, Sandflat, Gran Salitral, Pavocosa sp.

Funding: Agencia Nacional de Investigaciones Científicas y Tecnológicas 2013-1129 CONICET PIP 2014-2016 11220130100005CO Universidad Nacional de La Pampa PI09G This research was supported by projects PICT 2013-1129 (from Agencia Nacional de Investigaciones Científicas y Tecnológicas); PIP 2014-2016 11220130100005CO (from CONICET), and project PI09G (from Universidad Nacional de La Pampa) to Ricardo N Melchor. The funders had no role in study design, data collection and analysis, decision to publish, or preparation of the manuscript.

==============================
Pavocosa sp. (Lycosidae) burrows found in an open sparsely vegetated area on the edge of the Gran Salitral saline lake, in central Argentina, are described. Burrows were studied by capturing the occupant and casting them with dental plaster. The hosting sediments and vegetation were also characterized. Inhabited Pavocosa sp. burrows display distinctive features as open, cylindrical, nearly vertical, silk lined shafts about 120 mm long, subcircular entrances, a gradual downward widening, and a particularly distinctive surface ornamentation in the form of sets of two linear parallel marks at a high angle to the burrow axis. Instead, casts of vacated Pavocosa sp. burrows showed some disturbances caused either by the reoccupation by another organism or by predation of the dweller. Two morphologies are related to reoccupation of burrows: those with a structure in form of an “umbrella” and another with smaller excavations at the bottom of the burrow. Predation by small mammals produces funnel-shaped burrows. Both active and abandoned Pavocosa sp. burrow casts are compared with existing ichnogenera and inorganic sedimentary structures, highlighting its distinction. It is argued that key features like the presence of a neck, a downward widening and the described surface texture will allow recognition of wolf spider burrows in the fossil record. However, the putative spider burrows described in the literature either lack the necessary preservational quality or do not show ornamentation similar to the modern wolf spider burrows. Fossil wolf spiders are recorded since the Paleogene (possibly Late Cretaceous), therefore Cenozoic continental rocks can contain wolf spider burrows awaiting recognition. In addition, the particular distribution of Pavocosa sp. in saline lakes may imply that this type of burrow is linked to saline environments.

Introduction

Araneae (recorded since the Devonian) is the most diverse order within arachnids with around 47,000 described extant species (World Spider Catalog, 2017). Due to striking adaptations such as silk production and a complex behavior (e.g., construction of hunting webs), Araneae has become a highly successful group that is present in almost all environments (Murphy et al., 2006; Garrison et al., 2016). Burrow construction in spiders is considered a primary adaptation as a retreat from high temperatures and dry air conditions typical of arid environments (e.g., Cloudsley-Thompson, 1983; Punzo, 2000). Important functions as dwelling, nesting, mating, breeding, and foraging are also related to burrows (e.g., Marshall, 1996; Aisenberg, Viera & Costa, 2007; Hils & Hembree, 2015; Uchman, Vrenozi & Muceku, 2017).

In general, modern spider burrows consist of vertical or oblique, simple or branched forms, sometimes with a terminal chamber, in some cases silk lined, and structures atop such as trap doors or a turret can be found (e.g., Ratcliffe & Fagerstrom, 1980; Bryson, 1939; Hils & Hembree, 2015; Uchman, Vrenozi & Muceku, 2017). Among the burrowing spiders, those of the wolf spider (Lycosidae) tend to produce a nearly vertical burrow with or without a terminal chamber in flat terrain, whereas many trapdoor spider burrows (families Nemesiidae, Ctenizidae, Antrodiaetidae) are at an oblique angle and located on inclined surfaces (Uchman, Vrenozi & Muceku, 2017). This simple morphology can be comparable to the ichnogenenera Skolithos Hadelman, 1840 or Cylindricum Linck, 1949 (Smith et al., 2008; Hils & Hembree, 2015), the Y- shaped forms to Psilonichnus Fürsich, 1981 (Uchman, Vrenozi & Muceku, 2017), and those with a terminal chamber to Macanopsis Macsotay, 1967 (Hasiotis, 2002; Mikuś & Uchman, 2012; Hils & Hembree, 2015; Uchman, Vrenozi & Muceku, 2017).

Significant research related to burrow construction in wolf spiders has been made, but mainly focused on biological and ecological aspects (e.g., Hancock, 1899; Marshall, 1996; Aisenberg, Viera & Costa, 2007; Carrel, 2008; Suter, Stratton & Miller, 2011; De Simone, Aisenberg & Peretti, 2015; Foelix et al., 2016; Foelix et al., 2017; Framenau & Hudson, 2017). In addition to the pioneer contributions by Bryson (1939), Ahlbrandt, Andrews & Gwynne (1978), and Ratcliffe & Fagerstrom (1980), recent neoichnological studies have paid attention to the morphology of spider burrows (Hils & Hembree, 2015; Hembree, 2017; Uchman, Vrenozi & Muceku, 2017). These studies rely essentially on the overall morphology as a clue for recognition of spider burrows in general, including those of Lycosidae.

Similarly, probable spider burrows in the fossil record are scarce and their identification was always based on general morphology. The oldest record is controversial and based on poorly preserved simple vertical hollows from the Eocene of northern France, first considered worm burrows (Polychaeta) and later assigned to trapdoor spiders, in both cases named using biological names for a trace fossil (see details in Dunlop & Braddy, 2011). The same material was later incorrectly referred to as Oichnus (Bromley, 1981) by Dunlop & Braddy (2011), an ichnogenus reserved for bioerosion structures on calcareous skeletons (Wisshak et al., 2015). Skolithos isp. 1 from the Mio-Pliocene fluvial sediments of Brazil was compared with Lycosidae burrows due to its overall morphology (Fernandes, Borghi & Carvalho, 1992). Pleistocene and Holocene carbonate eolianites from the Bahamas and Yucatán contain Skolithos linearis (Haldeman, 1840) that were tentatively assigned to arachnids and/or insects (White & Curran, 1988; Curran & White, 1991; Curran & White, 2001). Finally, a burrow in Pleistocene clastic sediments of the Simpson Desert in Australia (Hasiotis, 2007) was attributed to wolf spiders.

The purposes of this work are (1) the identification of ichnological signatures of the burrows produced by Pavocosa sp. (Lycosidae) that may facilitate identification of wolf spider burrows in the fossil record, and (2) to discuss their environmental distribution.

Figure 1 Compilation of previous descriptions of wolf spider burrows.

(A) Geolycosa domifex (Hancock, 1899; fig. Pl II). (B) Generalized shape of spider burrows (Ctenizidae, Antrodiaetidae, Theraphosidae and Lycosidae. Ratcliffe & Fagerstrom (1980, fig. 1B). Not to scale. (C) Geolycosa xera archboldi and (D) G. hubbelli burrows by Carrel (2008), fig. 1). (E) Geolycosa missouriensis burrow (Suter, Stratton & Miller, 2011, fig. 1). (F) Geolycosa sp. (Chikhale, Santape & Bodkhe, 2013, fig. 7); (G) Allocosa brasiliensis: Produced by: a. Females, b. Males, and c. Juveniles (Albín, Simó & Aisenberg, 2015, fig. 1). (H) Hogna lenta: a. vertical shaft (fig. 12-2), b. vertical shaft with a terminal chamber (14-4), c. subvertical shaft (fig. 13-4), and d. Y-shaped burrow (fig. 15-1) (Hils & Hembree, 2015) (I) Tetralycosa burrow showing (a) offset burrow and (b) with original burrow backfilled ((Framenau & Hudson, 2017), fig. 3). (J) Allocosa senex (Foelix et al., 2017; fig. 16); (K) Trochosa hispanica (Uchman, Vrenozi & Muceku, 2017; fig. 6A). Image credit: Fatima Mendoza-Belmontes.

Previous descriptions of modern wolf spider burrows

The first work unequivocally related to burrows of wolf spiders was “The castle –building spider” from Illinois (USA) published by Hancock (1899). This paper describes in detail the burrows produced by Geolycosa domifex (Hancock, 1899) (= Lycosa domifex), explaining important aspects such as materials and the methods of construction. Geolycosa domifex burrows are described as vertical shafts, unless obstacles cause some deviation (Fig. 1A). Ratcliffe & Fagerstrom (1980), in their widely cited work on traces found in Holocene floodplains, described spider burrows in general (assigned to Ctenizidae, Antrodiaetidae, Theraphosidae and Lycosidae) as simple or branched tunnels, sometimes with side chambers that are separated from the main tunnel by hinged doors (Fig. 1B). Burrows of Geolycosa xera archboldi (McCrone, 1963) and G. hubbelli (Wallace, 1942) from Florida, USA, are illustrated as vertical shafts showing a gradual transition between the shaft and the terminal chamber (Figs. 1C–1D) (Carrel, 2008). Geolycosa missouriensis (Banks, 1895) burrows from Mississippi, USA, are described as vertical forms, narrower at the surface and broader near the bottom, sometimes with a conspicuously enlarged chamber at the bottom (Fig. 1E) (Suter, Stratton & Miller, 2011). Geolycosa sp. burrows from India, exhibited a contrasting morphology in comparison with previous records of wolf spiders. These burrows were complex with a U-shaped form, two chambers (one located at the entrance and the other at the end of the burrow), and shallow hollows described as drainages or prey traps (Fig. 1F) (Chikhale, Santape & Bodkhe, 2013). Albín, Simó & Aisenberg (2015), reported different burrow morphologies produced by Allocosa brasiliensis (Petrunkevitch, 1910) from Uruguay, linking these variations in the morphology to the development stage and sex of the spider that produce them. These authors described burrows with a simple vertical shaft and a terminal chamber produced by adults, shallow capsules by virgin females, and Y-shaped burrows by male juveniles (Fig. 1G). Hils & Hembree (2015), through experimental neoichnological studies, recorded four burrow morphologies produced by Hogna lenta (Hentz, 1844) (Lycosidae): vertical shafts, vertical shafts with a terminal chamber, sub-vertical shafts, and Y-shaped burrows (Fig. 1H). Geolycosa vultuosa (Koch, 1838) burrows from Albania are characterized as vertical to subvertical, slightly curved or straight shafts with a basal chamber, showing either a gradual transition between the shaft and the basal chamber or a well delineated chamber (Vrenozi & Uchman, 2015). In a taxonomic revision of the halotolerant wolf spider genus Tetralycosa Roewer, 1960 (Framenau & Hudson, 2017), the burrows of three species (T. alteripa McKay, 1976, T. williamsi Framenau & Hudson, 2017, and T. eyrei Hickman, 1944) were described. Tetralycosa burrows are vertical shafts with an offset (a curvature) at mid-depth, which are later modified by backfilling the part above the curvature and creating a new burrow oriented in the opposite direction (Fig. 1I) (Framenau & Hudson, 2017). Allocosa senex (Mello-Leitão, 1945) burrows from Uruguay are also simple vertical shafts with a downward widening (Fig. 1J) (Foelix et al., 2017). Finally, the burrows of Trochosa hispanica (Simon, 1870) from Albania (Fig. 1K) were described as simple, vertical shafts with a terminal chamber (Uchman, Vrenozi & Muceku, 2017).

From the previous account, it is clear that the most common form in wolf-spider burrow are almost vertical cylinders with a rounded end that increase progressively in width downward, vertical shafts with a terminal chamber, and Y shaped burrows. Hasiotis & Bourke (2006) also suggested that horizontal burrows systems with a pustulose ornamentation are produced by spiders, however, the illustrated burrow system (Hasiotis, 2002, p. 114, figure B) is typical of surface burrows produced by Grillotalpidae (e.g., Chamberlain, 1975). Figure 1 also highlights that the burrows produced under experimental conditions (Fig. 1H) contrast markedly with the remaining ones excavated in natural conditions.

Materials and Methods

We studied burrows produced by Pavocosa sp. found on the edge of sparsely vegetated sandflats of the Gran Salitral saline lake located in southwest La Pampa Province, Argentina (37°24′18.40″S, 67°12′13.57″W) (Figs. 2A–2B). This saline lake is placed in the subregion of alluvial plains of the Atuel-Salado rivers, characterized by a flat relief and sandy sediments, under a semiarid climate and with halophyte vegetation (Fig. 2C) (INTA, La Pampa, UNLPam, 1980). The Gran Salitral saline lake is the terminal part of an endorheic drainage system that occasionally receives waters from the Atuel- Salado rivers. Modern brines exhibit a concentration ranging from 213 to 252 g/l and the near-surface sediments of the saline lake attest for hydrological variations during the Holocene, including fluctuations in brine salinity and lake level (Melchor & Casadío, 2000). The mean monthly temperature ranges between 6.9 °C in July and 24.6 °C in January, and the mean annual precipitation is 340 mm, in both cases for the period 1961–1980 (INTA, La Pampa, UNLPam, 1980).

Figure 2 Location map of the study area.

(A–B) Site of study in the “Gran Salitral” in La Pampa Province, Argentina; (C) Geomorphologic map of the Gran Salitral area and location of Pavocosa sp. burrows (GS). Modified from Melchor et al. (2012). Image credit: Ricardo Melchor.

Figure 3 Measures taken on burrows.

Length (L), neck length (NL), minimum (mD) and maximum diameter (MD), angle of inclination (A). Image credit: Fatima Mendoza-Belmontes.

Observations were conducted during three field trips in October 2016 (early spring, mean monthly temperature for 2016: 15.4 °C, and the total monthly precipitation was 140 mm), December 2017 (late spring, mean monthly temperature for 2016: 23.1 °C, with no precipitations) and February 2017 (summer, mean monthly temperature for 2017: 24.7 °C, and precipitation was 22 mm). Rain data were obtained from Policía de la Provincia de la Pampa (2017) (http://www.policia.lapampa.gov.ar/contenidos/ver/lluvias), and temperature data from Servicio Meteorológico Nacional (2017) (http://www.smn.gov.ar), in both cases for the nearby 25 de Mayo and Puelén towns.

Sandflat sediments were logged in a shallow pit using standard sedimentological methods, and samples were taken for grain size and carbonate content analysis. Carbonate content of sediment samples was estimated using the Digital Calcimeter “NETTO” that indicates the total percent amount of calcium and magnesium carbonates. Grain size analyses of sediment samples were obtained by the laser particle size counter Malvern Mastersizer 2000® (Malvern, UK), prior to elimination of organic matter and carbonates, at the Laboratorio de Sedimentología of the Facultad de Ciencias Exactas y Naturales, Universidad Nacional de La Pampa.

A total of nine burrows were casted using dental plaster and three spiders found inside the burrows were collected for identification. Measurements on casts taken were the total length (L), neck length (NL), the minimum (mD) and maximum diameter (MD), and the angle of inclination (A). The measures on the sets of surface ridges preserved on the casts, were the length, the width, and the orientation in relation to the principal axis of the burrow (See Fig. 3). We also measured the entrance diameter (ED) from field photographs.

A 3D model of the burrows was generated based on photographs taken with a Lumix DMC-FZ70 camera (Panasonic, Osaka, Japan) and processed in the software Agisoft Photoscan Professional v.1.4.6. The resulting models were exported in OBJ files to Adobe Photoshop CC 2017 (TM) and converted to U3D files (a standard format for 3D), to compose a PDF file for easier visualization.

The casts and spider specimens collected were stored in the “Colección Paleontológica de la Facultad de Ciencias Exactas y Naturales” of the Universidad Nacional de La Pampa (acronym GHUNLPam), and one of the Pavocosa sp. specimens in the Museo Argentino de Ciencias Naturales “Bernardino Rivadavia” (acronym MACN-Ar). The specimens were preserved in EtOH 80%; photographs of preserved specimens were taken with a Leica DFC 290 digital camera mounted on a Leica M165 C stereoscopic microscope (Wetzlar, Germany). Images taken in different focal planes were combined with Helicon Focus 4.62 Pro (http://www.heliconsoft.com). The width between the fangs of chelicera in collected spider specimens was measured for comparison with the marks preserved in the casts.

Results

Ocurrence of Pavocosa sp. burrows

In early spring (October 2016) abundant burrow entrances of similar size were observed in the sandflat surface. Spider burrows were found in a sparsely vegetated sandflat (0 to 10% of plant coverage), with the only presence of the small halophyte shrub Heterostachys ritteriana Ungern-Sternberg, 1876 (Fig. 4A). The burrows were simple vertical and silk lined forms (Fig. 4B), appearing either open and covered with a thin ring of silk (Fig. 4C) or partially closed with a plug of silk and sediment pellets (Fig. 4D). Surrounding the burrow (in a radius of up to 64 cm) abundant small spherical sediment pellets were observed (with a density of up to 290 pellets/m2) (Fig. 4F), and at this time no casts were made. In late spring (December 2016) burrow density was lower, and they were restricted to a small area on the edge of the saline lake with sparse vegetation at the boundary with the bare sandflat. A total of eight casts were obtained, five were inhabited burrows, while the remaining were abandoned. The inhabited burrows showed up to two sacs of eggs in the lowermost part (Fig. 4E). During the field trip conducted in summer (February 2017), very few burrows were observed, all open and partially filled with some sand, and they seemed to be uninhabited for a long time. At this time only one uninhabited burrow was casted.

Figure 4 View of Pavocosa sp. burrows in the field and location of the study area.

(A) Site of observation of burrows in an open area with sparse vegetation (Heterostachys ritteriana). (B) Longitudinal section of an inhabited burrow with silk lining. Scale divisions in centimeters. (C) Entrance covered with a thin layer of silk. (D) Burrow partially closed with a cap of silk and sediment pellets; (E) Sac of eggs found inside the burrow. Scale divisions in millimetres. (F) Partially plugged entrance and sediment pellets dispersed on the surface of the sandflat. Photo and image credit: Ricardo Melchor and Fatima Mendoza-Belmontes.

Sandflat sediments

The pit dug in the saline sandflat where the burrows occur was 60 cm deep (Fig. 5A). The uppermost bed (# 1) is 13 cm thick and mainly composed of poorly-sorted pale yellowish brown (10 YR 6/2) silty sand containing 0.9% of carbonate (Figs. 5B, 5C). The lower 5 cm of bed 1 exhibits thin diffuse evaporite laminae and a mud lamina. This bed contained the studied Pavocosa sp. burrows. Bed 2 (7 cm thick) is poorly-sorted moderate yellowish brown (10 YR 5/4) silty sand, with massive structure and 0.8% of carbonate. Bed 3 (5 cm thick) is very poorly-sorted, dark yellowish brown (10 YR 4/2), silty sand with massive structure, containing 1.4% of carbonate and small (2 mm) gastropod shells comparable with Heleobia (Stimpson, 1865). The 27 cm thick bed 4 is very poorly-sorted, massive, moderate brown (5 YR 4/4), sandy silt containing 0.6% CO3. The 6 cm thick lowermost bed (# 5), is mainly composed of fine-grained, pale yellowish brown (10 YR 6/2) sand with abundant carbonate cement that matches with the water table. Field work was conducted in rainy days, however, the water table was well below the bottom of Pavocosa sp. burrows (about 40–45 cm below the bottom of the burrows).

Figure 5 Sediments of the sandflat.

(A) Detailed section of the sediments observed at the pit, wt, water table. (B) Representative grain size distribution of sediment samples. (C) Classification of sediment samples after Shepard (1954). Image credit: Ricardo Melchor and Fatima Mendoza-Belmontes.

Figure 6 Comparison between type material of Pavocosa gallopavo and Pavocosa sp.

(A) Female epigyne of Pavocosa gallopavo (MACN-Ar 13208), arrow pointing deep furrows on the atrium. (B) Female epigyne of Pavocosa sp. (MACN-Ar 38582), arrow pointing deep furrows on the atrium. (C) Dorsal view of Pavocosa gallopavo (MACN-Ar 13208). (D) Dorsal view of Pavocosa sp. (MACN-Ar 38582). Image credit: Luis Piacentini.

Producer of the burrows: Pavocosa sp.

Although the genus Pavocosa (Roewer, 1960) was never reviewed, and its composition was recently questioned (Toscano-Gadea & Costa, 2016), the inclusion of the material studied as an undescribed species of Pavocosa was possible through the comparison of the males and females of Pavocosa gallopavo (Mello-Leitão, 1941) (Figs. 6A, 6C), the type species of the genus. The male holotype of P. gallopavo (MLP-15065) and females from MACN collection were examined and they share with Pavocosa sp. (Figs. 6B, 6D) the presence of deep furrows on the atrium, parallel to the median septum of the female epigyne and the coloration pattern (Figs. 6A, 6B), characteristics probably diagnostic of the genus (L Piacentini, personal observations). The enlarged posterior eyes in Pavocosa sp. and the shape of the genitalia are clearly distinctive from P. gallopavo. The fangs of specimens captured inside the burrows (n = 3) are separated about 3.9–4.6 mm (Fig. 7H). The environmental distribution of Pavocosa is little known, although it seems to prefer bare patches in sandy grassland soils (L. Piacentini, pers. obs., 2014–2017).

Figure 7 Plaster casts of Pavocosa sp. burrows and details of surface ornamentation.

(A) GHUNLPam-4771. Dweller captured Pavocosa sp. (GHUNLPam -4780). (B) GHUNLPam -4772 (C) GHUNLPam -4773. Dweller captured. Pavocosa sp and an egg sac found at the bottom (GHUNLPam -4770). (D) GHUNLPam -4774. Egg sac found at the bottom (E) GHUNLPam -4775. (F–G) Surface texture of burrow casts in the form of sets of two linear parallel ridges (arrows) (H) View of cheliceral fangs of Pavocosa sp. (specimen GHUNLPam -4780). Photo and image credit: Fatima Mendoza-Belmontes.

Additional material of the described species from Córdoba (Salinas Grandes, 29°50′39″S, 64°40′16″W), Santiago del Estero and San Luis (Pampa de las Salinas; 32°12′19″S, 64°39′13″W) were recorded from MACN-Ar collection (23503, 23505 to 23513, 24096, and 38710), all from saline environments. The burrows of Pavocosa sp. from Córdoba (A Peretti, C Mattoni and M Izquierdo, pers. comm., 2008) and San Luis (M Ramírez, pers. comm., 2016) are very similar to those described in this work.

Pavocosa sp. burrows

The inhabited burrows (n = 5) (Figs. 7A–7E) are simple, vertical and circular shafts with an inclination of the main axis of 72°–88°(average: 80°), the length ranges from 115 to 130 mm (average: 120 mm). The diameter gradually increases from an upper narrow neck that is 12 to 15 mm wide (average 14 mm) and 5–8 mm long (average 6 mm), to a maximum diameter in the lower half ranging from 18 to 28 mm (average 23 mm). The outline of the entrance and cross-section of the maximum diameter of the burrows are subcircular. In average, the widest part of the burrow is 64% larger than the neck. The burrow cast surface of five burrow casts exhibits sparse ornamentation in the form sets of two linear parallel ridges (Figs. 7F–7G) about 2.8–4.4 mm long (average 3.4 mm, n = 16) and 2.2–4.5 mm wide (average: 3.4 mm, n = 14) aligned oblique to perpendicular (range: 42°–89°, average: 64°, n = 14) to the main axis of the burrow. The Supplemental Information contains interactive PDF files of each of the Pavocosa sp. burrow casts.

Figure 8 Plaster casts of modified Pavocosa sp. burrows.

(A–B) Burrows with umbrella-like structures in the middle part, probably produced by reoccupation by ants (GHUNLPam-4776 and 4777). (C–D) Plan view showing umbrella shape from burrow casts GHUNLPam-4776 and 4777. (E) Detail of the knobby surface texture of the umbrella-like structure. (F) Cast showing two smaller burrows arising from the bottom of the wolf spider burrow (GHUNLPam -4778). (G) Funnel-shaped burrow cast as result of predation by a small armadillo (GHUNLPam -4779). Arrows point to set of two parallel ridges. (H) Detail of the set of two linear parallel ridges (arrows). (I) Field view of burrow modified by predation by armadillos (cast figured in G). Note brecciated fragments produced during excavation by the armadillo. Photo and image credit: Ricardo Melchor and Fatima Mendoza-Belmontes.

Modified Pavocosa sp. burrows

Uninhabited Pavocosa sp. burrows (n = 4) display some kind of modification in their overall form (Figs. 8A, 8B, 8F, 8G) (see Supplemental Information for interactive 3D models of each cast). All are composed of a highly inclined shaft (range: 78°–87°; average: 84.5°), with an upper constriction and an average maximum diameter ranging from 15 to 22 mm (average 19 mm). Three types of modifications were identified. (1) Subcylindrical burrows (108–116 mm long by 15–22 mm wide) with a subhorizontal expansion in the middle part forming an ”umbrella” (Figs. 8A–8B). The shaft boundary exhibit scarce ornamentation in the form of sets of two linear parallel ridges similar to those of the inhabited Pavocosa sp. burrows. The “umbrella” structure shows an oval to lobed shape in the plan view (Figs. 8C–8D), with a minimum diameter of 47–54 mm and a maximum diameter of 59–66 mm. The “umbrella” surface exhibits an ornamentation in form of small (1.4 mm in diameter) rounded knobs (Fig. 8E). The burrow bottom is rounded or partially filled with sediments. (2) Subcylindrical burrow about 116 mm long and 21 mm wide with two smaller burrows (8 mm of diameter) arising at the bottom of the larger burrow (Fig. 8F). (3) A third form is a 143 mm high and 101 mm wide funnel that ends in a 24 mm wide cylindrical shaft with an oblique bottom (Fig. 8G). The surface of the funnel exhibits sets of two parallel ridges (about 21 mm long and 9.2 mm wide) running oblique to the major axis (Fig. 8H).

Discussion

Clues for identification of wolf-spider burrows in the fossil record

Pavocosa sp. produces open burrows with distinctive features as cylindrical, nearly vertical, silk lined shaft showing a gradual downward widening, a neck in the top and a rounded end, the entrance sometimes plugged with a cap of silk and sediment pellets, and a particularly distinctive surface ornamentation on the burrow margin. Most of these features are shared with other wolf spider burrows documented in the literature (Fig. 1) (Hancock, 1899; Ratcliffe & Fagerstrom, 1980; Carrel, 2008; Suter, Stratton & Miller, 2011; Albín, Simó & Aisenberg, 2015; Hils & Hembree, 2015; Vrenozi & Uchman, 2015; Foelix et al., 2017; Uchman, Vrenozi & Muceku, 2017). In particular, the presence of a neck and downward widening seem to be a common feature in wolf spider burrows found in natural settings. For Pavocosa sp. burrows this widening is about 64%, whereas it is 52% for Trochosa hispanica (Uchman, Vrenozi & Muceku, 2017).

Another highly distinctive feature of Pavocosa sp. burrows is their surface ornamentation in the form of two short parallel ridges oblique to perpendicular with the burrow axis that appear in most burrow casts (Figs. 7F–7G). Although this surface ornamentation was not recorded in some casts, probably due to the presence of the silk lining, all the burrow casts with delicate preservation of the surface texture exhibit these paired ridges. This feature was not identified in previous studies of wolf spider burrows and is potentially related to the burrowing technique used by Pavocosa sp. Spiders use two main mechanisms of excavation: (1) by pushing and compressing sediment using the pedipalps (Hils & Hembree, 2015) and (2) by scraping the soil with help of fangs from chelicerae (Stokes, 1884; Suter, Stratton & Miller, 2011; Hils & Hembree, 2015; Foelix et al., 2016). Although we have not observed Pavocosa sp. during digging, the sets of two linear parallel ridges observed on the surface of the better preserved burrow casts are similar in form and shape with the arrangement of fangs of collected specimens. The distance between fangs (3.9–4.6 mm) overlaps with the distance between ridges within a set (2.2–4.5 mm). Thus we propose that excavation in Pavocosa sp. involves the use of fangs, as in the type 2 excavation mechanism mentioned above.

Silk lined burrows are unique in spiders and essentially impart stability in soft substrates to prevent collapse (Ratcliffe & Fagerstrom, 1980; Foelix et al., 2017; Hils & Hembree, 2015). The presence of organic matter in the form of a silk lining increase the potential of preservation of wolf spider burrows (Uchman, Vrenozi & Muceku, 2017), well above those of all others arthropods that in habit the same environment.

Spider burrows may be modified by reoccupation or predation, as well as by environmental changes. Reoccupation of abandoned lycosid and mygalomorph burrows by lizards, centipedes, moths, wasps, beetles and ants have been documented (e.g., Fellows, Fenner & Bull, 2009). Ants have been also observed invading occupied wolf spider burrows with the purpose of prey piracy (Marshall, 1995). However, it has not been documented if the reoccupation results in any change in the morphology of the burrow. Common spider burrow disturbances caused by predation includes those produced by pompilid wasps that prey the spider and occasionally dig a tunnel perpendicular to the spider burrow (Gwynne, 1979; Costa, Pérez-Miles & Mignone, 2004), and excavation of the upper part of the burrows by armadillos (Suter, Stratton & Miller, 2011).

Most of Pavocosa sp. burrows are susceptible to go through a large amount of disturbances, including those caused by the reoccupation by another organism (Figs. 8A–8B and 8F) and predation of the dweller (Fig. 8G). Two kinds of burrow modifications observed during this study are tentatively related to reoccupation of burrows: those with an expansion in the middle part as a kind of “umbrella” (Figs. 8A–8B) and those with smaller excavations at the bottom of the burrow (Fig. 8F). Even if we cannot discard an inorganic origin (i.e., evaporite leaching) for the “umbrella” structure seen in some casts, it is highly reminiscent of oval to lobed ant nest chambers (Tschinkel, 2003). Although no ants were recorded when making the casts, they were commonly seen in the sandflat surface constructing nests within vertebrate footprints and abandoned burrows, presumably to avoid the hard efflorescent salt crust of the sandflat surface. The producer of the smaller burrows at the bottom of Pavocosa sp. burrow is unknown. Funnel shaped burrows (Fig. 8G) are similar to the probing marks related to predation by small mammals, and similar structures are described in the literature including Sarzetti & Genise (2011) from northern Argentina, Suter, Stratton & Miller (2011): Fig. 2), and Platt (2014), the latter two from Mississippi, USA. Small mammals found in this area with similar behaviours are the armadillos and skunks. The more likely producer is a small armadillo as suggested by the size of the funnel and most importantly by the presence of sets of two large ridges in the cast surface (compare Platt, 2014), interpreted as scratch marks (Fig. 8H).

Preservation of burrows in the margin of saline lakes, including those of wolf spiders, is affected by environmental factors like early cementation by evaporites and swelling of expansive clays during flooding (e.g., Scott, Renaut & Owen, 2010). Cementation by evaporites favors preservation, whereas wetting and drying cycles of swelling clays can destroy the burrows.

Both the original Pavocosa sp. burrows and those modified by reoccupation or predation can be compared with known ichnogenera. The simple vertical forms are grossly comparable with Skolithos (see Alpert, 1974; Schlirf, 2000); some significant differences are the presence of a constriction or neck, the downward widening and the surface texture. These features are potentially significant ichnotaxonomicaly (Schlirf & Uchman, 2005), although no proposed ichnotaxon match them. Slight variations in burrow diameter are allowed in Skolithos (Alpert, 1974; Schlirf, 2000), although the observed differences in Pavocosa sp. burrow diameter are significant and repetitive. There are a few examples of ornamented Skolithos, all of them from continental settings and tentatively assigned to insects or spiders, but they are not comparable to that observed in Pavocosa sp. burrows (Bromley & Asgaard, 1979; Schlirf, Uchman & Kümmel, 2001; Netto, 2007). These ornamented Skolithos exhibit indistinct striations, except for the example described by Netto (2007) that display horizontal striae forming a circular ring. In consequence, there is no known fossil burrow with all the features described for the studied wolf spider burrows.

Modified Pavocosa sp. burrows with an “umbrella” if fossilized can be confused with Daimoniobarax (Smith et al., 2011); in particular, the umbrella is comparable with chambers and the vertical burrow of the spider is comparable with the shaft connecting the chambers in Daimoniobarax. A potential difference is the considerably larger diameter of the burrow connecting the chambers that averages 40% of chamber diameter in the modified Pavocosa sp. burrow and 10% in Daimoniobarax (Smith et al., 2011). The modified Pavocosa sp. burrow with smaller burrows arising from the bottom can be confused with a downward bifurcation as seen in rhizoliths (Klappa, 1980), a roughly similar rhizolith was figured by Melchor, Genise & Miquel (2002), Fig. 3B). Finally, funnel shaped burrows can be compared with several ichnogenera including Monocraterion (Torell, 1870); Conostichnus (Lesquereux, 1876); Rosselia (Dahmer, 1937); Conichnus (Männil, 1966); and Cornulatichnus (Carroll & Trewin, 1995) (see also Platt, 2014). A fundamental difference with these ichnogenera is the lack of large paired surface ridges, as seen in the predated Pavocosa sp. burrow. Further differences are: (1) Monocraterion shows smaller radial burrows arising from the central funnel (Jensen, 1997); (2) Conostichnus exhibits a duodecimal symmetry and transverse and longitudinal ridges and furrows (Pemberton, Frey & Bromley, 1988); (3) Rosselia is a bulbous structure with a concentrically laminated fill (Schlirf, Nara & Uchman, 2002); (4) Conichnus exhibits a rounded apex and common chevron-like fill (Pemberton, Frey & Bromley, 1988); and (5) Cornulatichnus has a well-developed lining (Carroll & Trewin, 1995). Conical sedimentary structures of inorganic origin can also resemble Pavocosa sp. burrows modified by predation. Buck & Goldring (2003) identified two main inorganic processes that produced conical sedimentary structures: collapse and dewatering. The former is distinguished by V or U shaped downwarping of lamination and the latter by deformed lamination and massive zone at the base of the cone (Buck & Goldring, 2003). These features allow distinction from the predated (i.e., funnel-shaped) Pavocosa sp. burrow, that would have a massive fill.

Burrowing spiders belong to Mesothelae and Opisthothelae (Coddington, 2005). Although Mesothelae dates back to the Late Carboniferous, the only known burrowing group (Liphistiidae) has no fossil record (Dunlop, Penney & Jekel, 2017). Within Opisthothelae, burrowing spiders are found in the Middle Triassic to Recent Mygalomorphae, which includes the tarantulas and trapdoor spiders, and in the Cretaceous to Recent Lycosoidea (included in Araneomorphae) that comprises the wolf spiders (Dunlop, 2010; Dunlop, Penney & Jekel, 2017). The oldest putative example of Lycosoidea comes from Turonian beds of Botswana (Selden, Anderson & Anderson, 2009), which is close to the age of the superfamily suggested by phylogenetic studies (70 Ma, after Garrison et al., 2016), although most fossil records are from the Paleogene to Recent (Dunlop, Penney & Jekel, 2017). In addition, phylogenetic studies on web type suggest that the spider common ancestor likely foraged from a subterranean burrow, mostly sealed by a trapdoor (Garrison et al., 2016). In consequence, the record of spider burrows can be traced back at least to the Middle Triassic (and probably to the Late Carboniferous) and lycosid burrows to the Late Cretaceous or Cenozoic.

The use of fossils to calibrate molecular phylogenies is an uprising topic in spider biology (Planas, Fernández-Montraveta & Ribera, 2013; Wood et al., 2013; Moradmand, Schönhofer & Jäger, 2014). The absence of reliable fossil record, such as in Lycosidae (Penney, 2001), is an important impediment and the potential identification of wolf spider burrows on the fossil record, with the clues provided herein, can be a useful alternative source of data.

Environmental distribution of Pavocosa sp. burrows

The sediments of the sandflat containing the Pavocosa sp. burrows reflect the interaction between the nearby eolian and lacustrine settings. The two upper beds are essentially sandy deposits with a mixture of dominant fine sand and silt (samples S1 and S2; Fig. 5). The dominance of the coarse fraction (fine sand), poor sorting and the frequency distribution are comparable with those of modern interdune deposits (e.g., Ahlbrandt, 1979). Poorly defined laminae with evaporites in bed 1 are interpreted as the result of capillary rise and precipitation from brines. The sandy nature of the material where Pavocosa sp. excavated the burrows and their location 40 cm above the water table suggests preference for well-drained substrates. In contrast, the lowermost silty beds (samples S3 and S4; Fig. 5) are interpreted as dominantly lacustrine deposits, on the basis of the fine grain size and the presence of gastropods shells. Heleobia is a very common extant gastropod in South America recorded in estuarine and continental settings, including saline lakes (see review in Cazzaniga, 2011). In consequence, the logged section reflects the migration of the parabolic dune towards the northeast over the Gran Salitral lacustrine sediments (for a more detailed interpretation of dune deposits see Melchor et al., 2012). The presence of abundant Pavocosa sp. burrows in the well-drained sandflat deposits of the Gran Salitral and similar occurrences reported in the literature (e.g., Hudson & Adams, 1996) suggest that wolf spider colonization of saline lakes occur preferentially during dry periods of the lake.

Wolf spiders (Lycosidae) are one of the most successful spider families distributed in most of the habitats around the world (World Spider Catalog, 2017). Lycosids display a wide range of prey-capture strategies from web builders to burrow-dwellers or vagrant species. The use of burrows in wolf spiders can be in some cases obligatory, temporary in male juveniles, and as brood care in females (Logunov, 2011), or merely facultative in absence of objects as a rock that serves as a retreat. In general, burrows in wolf spiders are related to open areas of xerothermic habitats with sparse or no vegetation (e.g., sandy seashores, dune heaths, limestone areas and desert nanophanerophyte steppe) (Logunov, 2011). Some wolf spider species have specific habit preferences, as is the case of halotolerant species that inhabit the surface of salt lakes, most of them included in Tetralycosa and other species as Lycosa salifodina (McKay, 1976) from Australia (Hudson & Adams, 1996; Framenau & Leung, 2013), and two other Argentinian species including Pavocosa sp. In particular, Pavocosa sp. has been documented in saline lakes of Cordoba, Santiago del Estero, San Luis and La Pampa. In consequence, it is likely that the described burrows are typical of saline environments.

Conclusions

Observations on the burrows of the wolf spider Pavocosa sp. in the coast of a saline lake in central Argentina suggest that:

(1) Pavocosa sp. produces burrows with recognizable features as open, cylindrical, nearly vertical, silk lined shafts, showing a gradual downward widening, with a neck and the entrance and a rounded end, and a particularly distinctive surface ornamentation on the burrow margin. These features are considered typical of wolf spider burrows.

(2) Burrows are susceptible to go through a large amount of disturbances, including reoccupation by another organism or by predation of the dweller. Two types of modified Pavocosa sp. are related to reoccupation of burrows: those with a lateral expansion in the middle part as a kind of ”umbrella” and another with smaller excavations at the bottom of the burrow. Predation by small mammals results in funnel-shaped burrows.

(3) Pavocosa sp. burrows have significant differences with those found in the Skolithos ichnospecies. Such features as the presence of a neck, a downward widening and the surface texture make them identifiable in the fossil record. The modified Pavocosa sp. burrows can be confused with Daimoniobarax, rhizoliths, and several conical sedimentary structures, although some key aspects allow their distinction.

(4) The features of Pavocosa sp. burrows that are considered diagnostic of wolf spider burrows are not identified to date in any published description of fossil examples.

(5) Pavocosa sp. colonized well drained sandy substrates of eolian origin on the margin of a saline lake. Known occurrences of this species suggest that it is a halotolerant wolf spider that inhabits the surface of saline lakes. Furthermore, as the wolf spiders avoid flooded substrates, it is suggested that the occurrence of wolf spider burrows in saline lakes is probably related to dry periods.

(6) The potential record of wolf spider burrows dates back to the Paleogene (possibly to the Late Cretaceous). The presence of silk lining increases its potential of preservation and the typical morphology and the surface texture render them recognizable in the fossil record.

Supplemental Information

Supplemental Information 1 Cast GHUNLPam-4771

Length = 131 mm; Neck Length = 8 mm; Minimum Diameter = 15 mm; Maximum Diameter = 22 mm; Angle = 84º. Dweller captured (Pavocosa sp GHUNLPam-4770). 3D model credit: Fatima Mendoza-Belmontes.

Click here for additional data file.

Supplemental Information 2 Cast GHUNLPam-4772

Length = 115 mm; Neck Length = 5 mm; Minimum Diameter = 15 mm; Maximum Diameter = 18 mm; Angle = 88º. 3D model credit: Fatima Mendoza-Belmontes.

Click here for additional data file.

Supplemental Information 3 Cast GHUNLPam-4773

Length = 130 mm; Neck Length = 6 mm; Minimum Diameter = 16 mm; Maximum Diameter = 27 mm; Angle = 75º. Dweller captured (Pavocosa sp GHUNLPam-4770). Eggs sac found on the bottom. 3D model credit: Fatima Mendoza-Belmontes.

Click here for additional data file.

Supplemental Information 4 Cast GHUNLPam-4774

Length = 106 mm; Neck Length = 6 mm; Minimum Diameter = 13 mm; Maximum Diameter = 23 mm; Angle = 72º. Sac of eggs found on the bottom. 3D model credit: Fatima Mendoza-Belmontes.

Click here for additional data file.

Supplemental Information 5 Cast GHUNLPam-4775

Length = 118 mm; Neck Length = 7 mm; Minimum Diameter = 13 mm; Maximum Diameter = 25 mm; Angle = 85º. 3D model credit: Fatima Mendoza-Belmontes.

Click here for additional data file.

Supplemental Information 6 Cast GHUNLPam-4776

Length = 111 mm; Minimum Diameter = 15 mm; Maximum Diameter = 15 mm; Angle = 86º; ”Umbrella” Structure: Diameter = 47 × 59 mm. 3D model credit: Fatima Mendoza-Belmontes.

Click here for additional data file.

Supplemental Information 7 Cast GHUNLPam-4777

Length = 106 mm; Minimum Diameter = 15 mm; Maximum Diameter = 22 mm; Angle = 78º.”Umbrella” Structure Diameter = 54 × 66 mm. 3D model credit: Fatima Mendoza-Belmontes.

Click here for additional data file.

Supplemental Information 8 Cast GHUNLPam-4779

Length = 130 mm; Minimum Diameter = 12 mm; Maximum Diameter = 100 mm; Angle = 85º. 3D model credit: Fatima Mendoza-Belmontes.

Click here for additional data file.

Supplemental Information 9 Cast GHUNLPam-4778

Cast GHUNLPam-4778. Length = 116 mm; Minimum Diameter = 15 mm; Maximum Diameter = 22 mm; Angle = 87º. Smaller burrows with around 33 mm length and 8 mm of diameter. 3D model credit: Fatima Mendoza-Belmontes.

Click here for additional data file.

Supplemental Information 10 Environmental distribution of the specimens from the MACN collection compared with Pavocosa sp

Click here for additional data file.

Silverio Feola, Mauricio Fernández, Sofía Mulatero, Luis Torres and Angélica Tamame helped during field work. The landowner of Puesto La Porfía, Mr Tránsito Cerda, is thanked for permission to work on his property.

Additional Information and Declarations

Competing Interests

Author Contributions

Data Availability

The authors declare there are no competing interests.

Fatima Mendoza Belmontes and Ricardo N. Melchor conceived and designed the experiments, performed the experiments, analyzed the data, contributed materials/analysis tools, prepared figures and/or tables, authored or reviewed drafts of the paper, approved the final draft.

Luis N. Piacentini prepared figures and/or tables, authored or reviewed drafts of the paper, approved the final draft.

The following information was supplied regarding data availability:

The casts and spider specimens collected were stored in the “Colección Paleontológica de la Facultad de Ciencias Exactas y Naturales” of the Universidad Nacional de La Pampa (acronym GHUNLPam 4770 -4780), and one of the Pavocosa sp. specimens in the Museo Argentino de Ciencias Naturales “Bernardino Rivadavia” (acronym MACN- Ar 38582). Additionally, the raw data are available as Supplementary Data Files.

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
