# Peer review of "Wolf spider burrows from a modern saline sandflat in central Argentina: morphology, taphonomy and clues for recognition of fossil examples"

_PeerJ, doi:10.7717/peerj.5054_

## Round 0.1 · original submission · Minor Revisions

Dear authors,

This is a very nice article, I enjoyed reading it. We have three review reports right now and they have suggested that this is an important contribution mainly because of its originality and should be published. One main concern shared by two reviewers and me is the disorganization and inconsistency of the reference list. Please, see the review reports with detail, and also my own annotated pdf, where I have marked in yellow the main inconsistencies. You can also revise the list using a PeerJ article already published for help. As an example, all the references should be written under the same guidelines; all use only a space after the name of the journal in italics, and there should not be a space before the page numbers.

Apart from that, Reviewer 3 made an interesting observation about the scarce development seen in your manuscript about the fossil spider burrows, which I agree with the reviewer, is an important subject of your work. It is true that you included a section for to deal with fossil burrows, but certainly, you describe very few of fossils in that section. Thus, I think that you have to pay attention to the Reviewer 3 observation and fix the manuscript in that aspect that will improve more yet your article. In this sense I think that the presence of the parallel ridges which, as you state, characterize all the Pavocosa sp. burrows, were no well shown in some figures. For instance, I cannot see the ridges on the plaster casts of your figures 7 (a-d) and 8 (a-b; f). Could you please add arrows to point them? Otherwise, if they would not be present in some fossil burrows, it means that it was not produced by Pavocosa sp? Other question that comes to my mind is if other taxon apart from Pavocosa sp. may construct the burrows using the fangs, you can apply other characters to solve the taxonomic determination with confidence? It is important to clarify this in the manuscript, and I think that a more detailed study of the fossils will help. The relevance of the application of these features in the recognition of Pavocosa sp. in ancient deposits will be allied to its influence in the construction of paleoenvironmental hypotheses.

I hope you find useful the comments and suggestions and I expect to see the revised version of your manuscript very soon.

Best regards,
Graciela Piñeiro

Reviewer 1 ·

Basic reporting

The submitted manuscript Wolf spider burrows from a modern saline sandflat in central Argentina: morphology, taphonomy and recognition of fossil examples
complies with the standards of the journal. In addition, it adjusts to the indicated
the “Editorial Criteria” of PeerJ (Basic Reporting, Experimental Design and Validity of the findings).

Experimental design

The submitted manuscript Wolf spider burrows from a modern saline sandflat in central Argentina: morphology, taphonomy and recognition of fossil examples
complies with the standards of the journal. In addition, it adjusts to the indicated
the “Editorial Criteria” of PeerJ (Basic Reporting, Experimental Design and Validity of the findings).

Validity of the findings

The submitted manuscript Wolf spider burrows from a modern saline sandflat in central Argentina: morphology, taphonomy and recognition of fossil examples
complies with the standards of the journal. In addition, it adjusts to the indicated
the “Editorial Criteria” of PeerJ (Basic Reporting, Experimental Design and Validity of the findings).

Additional comments

The article is a very good descriptive work of a theme little known worldwide and in particular in South America and presents an unusual perspective on the knowledge of the behavior and life habit of ancient spiders allowing to make parallels with the current species.

The work has a solid and updated bibliographic support.

I strongly suggest that the authors review the text in detail and conform to the characteristics of the journal.

Annotated reviews are not available for download in order to protect the identity of reviewers who chose to remain anonymous.

·

Basic reporting

This a good, orginal paper on rarely undertaken topic, i.e. on spider burrows from ichnological side. It contains new data and should be published. I encountered only some minor problems, which are indicated in the text. The refrences need some adjustment to the journal style

Alfred Uchman

Experimental design

The paper is based on standard field research, whcih are sufficient for ruch a paper

Validity of the findings

this one a a few papers dealing with spider burrows, with application ichnological approach

Additional comments

as above

Reviewer 3 ·

Basic reporting

The work is a description of wolf spiders burrows morphologies linked with ecological and soil data. The purpose of the work is 1) to compare those descriptions of burrows with fossils records and 2) discuss the ecology of wolf spiders’burrows.
I like this work because of the serious literature revision about modern spiders’ burrow, the precision and variety of the data, the complete discussion. Information about ecology of the spiders’ burrow is very interesting. I feel a little bit frustrated as I thing this work may be improved if giving more data about fossils information (using previous works) mostly because it is presented as the first goal of the study and is finally spotted only in the discussion and ever not mentioned in material and methods. I suggest author explain how the will generate and discuss or compare the information about comparison for fossil information to have a better integration of this goal. However I thing that the work just need this adjustment to integrate better the goal about fossils identification, and that the work deserve to be published.
I recommend this publication.

Summary
- Summary describe with clarity the observations, results and implications. I suggest at the beginning of the summary, add a sentence to introduce the further comparison with ichnogenera and understand in the beginning the focus of the work.
Introduction
- The introduction present a very well description about modern spiders’ burrows architecture/form.
- The introduction give well described information about wolf spider burrow in fossils records.
- The purpose of the work in clearly exposed.
- In the previous description of modern wolf spider burrows, authors give plenty and complete information about wolf spider burrow morphologies.

Material and methods
- The site in where burrows where observed and spiders captured is well described.
- The methodology for burrows descriptions is also well described.
- I was looking for information about comparison between those burrows and fossils to have an idea of how authors will respond to the principal focus of the work as explained in the introduction. Please give in material and method information necessary to understand how this question will be answer. Even if this is not part of the field work.
Results
- Line 180, the comment about rains looks important but what is the relation between rains and frequency of burrows entrance?
- Description of sandflat sediment is well done.
- The description of burrows is complete.
Discussion
- Discussion in complete and well written.

Experimental design

Complete and easy to read. I particularly like the precision and the complete approach of the burrows descriptions.

Validity of the findings

The finding is interesting as it is a whole approach and description of wolf spiders burrows. Also this results may have great implication for fossils identification.

Additional comments

I like this work because of the serious literature revision about modern spiders’ burrow, the precision and variety of the data, the complete discussion. Information about ecology of the spiders’ burrow is very interesting. I feel a little bit frustrated as I thing this work may be improved if giving more data about fossils information (using previous works) mostly because it is presented as the first goal of the study and is finally spotted only in the discussion and ever not mentioned in material and methods. I suggest author explain how the will generate and discuss or compare the information about comparison for fossil information to have a better integration of this goal. However I thing that the work just need this adjustment to integrate better the goal about fossils identification, and that the work deserve to be published.

---

## Round 0.2 · Minor Revisions

Dear authors,

I have read your revised version carefully and I realized that the title of your previous draft was indeed the responsible of Reviewer 3 and myself misinterpretation about the real subject of the article. I understand now that your study about the Recent spider burrows would be useful for comparative purposes of eventually collected fossil burrows in the future. But according to the antecedents that you presented about the bad preservation of these structures in the fossil record, it turns difficult the use of at least some of the described characters, excepting in depositional environments that favor exceptional preservation, even most delicate features. It is interesting that such kind of environments include conditions of high salinity and low energy (salt lakes) as those that members of Lycosidae prefer. Perhaps the reason for so bad preservation of the burrows is due to the fact that these spiders colonize the surface of the salt lakes at the drier most facet of the lake life (Hudson & Adams, 1996), being a very interesting aspect for paleoenvironmental reconstructions. Thus, I think that the taphonomic issue of the already found burrows (in a general view) and this last environmental evidence should be included in the discussion of your article to show the relevance of spider burrows identification in ancient deposits..

Ref: Hudson, P. and Adams, M. 1996. Allozyme characterisation of the Salt Lake spiders (Lycosa: Lycosidae: Araneae) of Southern Australia: Systematic and population genetic implications. Australian Journal of Geology, 44:535-567.

Best regard,
Graciela Piñeiro

---

## Round 0.3 · Minor Revisions

Dear authors,
I am glad to see that your manuscript is now almost ready to be accepted for publication in PeerJ. I realized that you found useful my last comments and incorporated them into the draft. I think that this will be a very useful contribution to the identification of wolf spider burrows in the future, because as you correctly pointed, researchers will have specific references for to make a correct identification. I have marked some remaining minor grammar corrections that you will be able to assess easily and quickly from the annotated pdf attached to this letter, so, I expect to see the revised version submitted very soon.

Best regards,
Graciela Piñeiro

---

## Round 0.4 · accepted · Accept

Dear authors,

I am very glad to let you know that your manuscript “Wolf spider burrows from a modern saline sandflat in central Argentina: morphology, taphonomy and clues for recognition of fossil examples” is now ready to be published in PeerJ. Congratulations!

With kind regards,
Graciela Piñeiro

#